# Methuosis Contributes to Jaspine-B-Induced Cell Death

**DOI:** 10.3390/ijms23137257

**Published:** 2022-06-29

**Authors:** Núria Bielsa, Mireia Casasampere, Jose Luis Abad, Carlos Enrich, Antonio Delgado, Gemma Fabriàs, Jose M. Lizcano, Josefina Casas

**Affiliations:** 1Research Unit on BioActive Molecules, Department of Biological Chemistry, Institute for Advanced Chemistry of Catalonia (IQAC-CSIC), 08034 Barcelona, Spain; nubielsa@gmail.com (N.B.); mireia.casasampere@iqac.csic.es (M.C.); joseluis.abad@iqac.csic.es (J.L.A.); gemma.fabrias@iqac.csic.es (G.F.); 2Unitat de Biologia Cel·lular, Departament de Biomedicina, Facultat de Medicina i Ciències de la Salut, Universitat de Barcelona, 08036 Barcelona, Spain; enrich@ub.edu; 3Centre de Recerca Biomèdica CELLEX, Institut d’Investigacions Biomèdiques August Pi i Sunyer (IDIBAPS), 08036 Barcelona, Spain; 4Unit of Pharmaceutical Chemistry (Associated Unit to CSIC), Department of Pharmacology, Toxicology and Medicinal Chemistry, Faculty of Pharmacy and Food Sciences, University of Barcelona, 08028 Barcelona, Spain; 5Liver and Digestive Diseases Networking Biomedical Research Centre (CIBEREHD), ISCIII, 28029 Madrid, Spain; 6Protein Kinases and Signal Transduction Laboratory, Departament de Bioquímica i Biologia Molecular and Institut de Neurociències, Universitat Autònoma de Barcelona (UAB), 08193 Barcelona, Spain; 7Protein Kinases in Cancer Research, Vall Hebron Institut de Recerca (VHIR), 08193 Barcelona, Spain

**Keywords:** autophagy, methuosis, apoptosis, sphingolipids, cytoplasmic vacuolization

## Abstract

Methuosis is a type of programmed cell death in which the cytoplasm is occupied by fluid-filled vacuoles that originate from macropinosomes (cytoplasmic vacuolation). A few molecules have been reported to behave as methuosis inducers in cancer cell lines. Jaspine B (JB) is a natural anhydrous sphingolipid (SL) derivative reported to induce cytoplasmic vacuolation and cytotoxicity in several cancer cell lines. Here, we have investigated the mechanism and signalling pathways involved in the cytotoxicity induced by the natural sphingolipid Jaspine B (JB) in lung adenocarcinoma A549 cells, which harbor the G12S K-Ras mutant. The effect of JB on inducing cytoplasmic vacuolation and modifying cell viability was determined in A549 cells, as well as in mouse embryonic fibroblasts (MEF) lacking either the autophagy-related gene *ATG5* or *BAX/BAK* genes. Apoptosis was analyzed by flow cytometry after annexin V/propidium iodide staining, in the presence and absence of z-VAD. Autophagy was monitored by LC3-II/GFP-LC3-II analysis, and autophagic flux experiments using protease inhibitors. Phase contrast, confocal, and transmission electron microscopy were used to monitor cytoplasmic vacuolation and the uptake of Lucifer yellow to assess macropinocyosis. We present evidence that cytoplasmic vacuolation and methuosis are involved in Jaspine B cytotoxicity over A549 cells and that activation of 5′ AMP-activated protein kinase (AMPK) could be involved in Jaspine-B-induced vacuolation, independently of the phosphatidylinositol 3-kinase/protein kinase B/mechanistic target of rapamycin complex 1 (PI3K/Akt/mTORC1) axis.

## 1. Introduction

In living organisms, there is a regulated equilibrium between the generation of new cells by cell division and the elimination of the damaged ones. The most well-known mechanism of programmed cell death is apoptosis [1]. In recent decades, however, new types of cell death independent of apoptosis have gained interest for their potential importance in pathological processes, toxicology, and cancer therapy [2]. One of the latest additions to the list of phenotypes of cell death is methuosis [3], first described by Overmeyer et al. in 2008 [4]. The name derives from the Greek *methuo*, which means “to drink to intoxication”, because in this type of cell death, the cytoplasm is occupied by fluid-filled vacuoles that originate from macropinosomes.

Macropinocytosis is a clathrin-independent endocytic process by which cells internalize extracellular fluid, nutrients, and proteins in vesicles (macropinosomes) generated from protrusion of the plasma membrane [2]. Macropinosomes enter the endocytic pathway and are recycled to the plasma membrane, or mature and become late endosomes that eventually fuse with lysosomes [5]. In methuosis, endosomal trafficking is not functional, and the vacuoles that are originated from macropinocytic activity are accumulated in the cytoplasm and they fuse with each other, producing larger vacuoles that finally occupy most of the cytoplasm. Cytoplasmic vacuolization can be transient or irreversible. Irreversible vacuolization produces different types of caspase-independent cell death, including methuosis, paraptosis, oncosis, and necroptosis [6], which occur without cellular shrinkage and nuclear fragmentation, typical of apoptosis [7]. The appearance of vacuoles can resemble autophagy but, in contrast to autophagosomes that have double membranes surrounding luminal cytoplasmic contents, methuosis vacuoles have a single membrane [7]. Paraptosis and oncosis proceed with cytoplasmic vacuoles, but in these cases, vacuoles are originated from endoplasmic reticulum or mitochondria [2] instead of macropinocytosis. 

Methuosis was originally described in glioblastoma cells after the ectopic expression of activated Ras (rat sarcoma virus) [4] and small molecules that induce methuosis in numerous cancer cell types have been described, including apoptosis-resistant cancer cells. Among these molecules, indol-based chalcones have been explored in U251 human glioblastoma cells [7,8,9], as well as the CK2 (casein kinase II) inhibitor silmitasertib in colorectal cancer cells [10].

Jaspine B (JB) is a natural anhydrous sphingolipid (SL) derivative isolated from the marine sponge *Jaspis sp*. reported to induce cytoplasmic vacuolation [11] and cytotoxicity in several cancer cell lines [11,12,13]. SL metabolism is a complex network of regulated reactions. SLs are important constitutive elements of the cellular membranes, and also display roles as bioactive molecules and intervene in different cellular-signalling pathways. JB interferes in SL metabolism acting as a competitive inhibitor of the CerS (ceramide synthase) enzymes [11]. Alterations in the SL metabolism have been associated with cell death by apoptosis and autophagy [12,13]. However, JB induces methuosis independently of the alterations it induces in the SL metabolism [11]. Importantly, JB capacity for inducing vacuolation is not cell-type-dependent, since it has been demonstrated in several cell lines with different genetic profiles [13]. Previously, we reported that JB induced cytoplasmic vacuolization in the human gastric cancer HGC27 cell line [11], which expresses wild-type K-RAS [14]. Since it has been reported that active K-Ras promotes methuosis [4,15], we investigated the activity of JB in lung adenocarcinoma A549 cells, which express the active G12S K-Ras mutant. Here, we describe that JB induces methuosis in these cells thorough activation of the 5’ AMP-activated protein kinase (AMPK).

## 2. Results

### 2.1. JB Induces Cytoplasmic Vacuolization in A549 and MCF7 Cells 

Lung adenocarcinoma A549 cells were treated with different concentrations of JB (Figure 1A) (CC_50_/24 h = 2.05 µM) (Appendix A) and analyzed for vacuolization at different times. In agreement with the results previously reported for human gastric cancer HGC27 cells [11], phase contrast microscopy showed that 5 µM JB induced the formation of cytosolic vacuoles (4 h) in A549 cells (Figure 1B), with a cellular viability of ~80%. These conditions were selected for vacuolization induction in further experiments.

JB interferes with SLs’ metabolism by inhibiting CerS [11]. To establish the role of CerS in cellular vacuolation induced by JB, we used Fumonisin B1 (FB1), a canonical CerS inhibitor [16]. As shown in Figure 1B, no vacuoles were observed in A549 cells treated with FB1, indicating that inhibition of CerS by JB is not implicated in its capacity to induce cytoplasmic vacuolation.

Transmission electron microscopy (TEM) revealed that JB-induced vesicles are multisized and surrounded by a single membrane (Figure 1C), in contrast to the double-membrane structures that characterize autophagosomes (see below). Interestingly, TEM showed the presence of lamellipodial membrane projections, suggesting that the vacuoles were originated from macropinocytic activity (Figure 1C). To investigate this possibility, we incubated cells with JB in the presence of the fluorescent dye Lucifer yellow (LY), a fluid-phase tracer that is incorporated in cells during macropinocytosis [4,17]. Fluorescence microscopy analysis showed round-shaped vacuoles filled with the fluorescent tracer LY, which matched with those observed by contrast phase imaging (Figure 1D). This indicates that vacuoles exhibited characteristics of macropinosomes, as reported for other cancer cell lines [11].

To confirm that JB-induced vacuolization occurred via macropinocytosis, we used EIPA (5-[*N*-ethyl-*N*-isopropyl]amiloride), a Na^+^/H^+^ exchanger inhibitor that blocks macropinocytosis [18,19]. The incubation of A549 cells with EIPA inhibited vacuolization induced by JB (Figure 1E). Furthermore, EIPA significantly rescued the viability of cells treated with JB (Figure 1F), indicating that vacuolization by macropinocytosis was implicated in JB cytotoxicity.

When the endocytic pathway is functional, vacuoles internalized by macropinocytosis are either recycled to the plasma membrane, or mature and become late endosomes that eventually fuse with lysosomes [20,21]. Despite the rise in the macropinocytic activity, methuosis induces a dysfunctional macropinocytotic process, so the nascent macropinosomes do not recycle or fuse with lysosomes [4]. To investigate whether the cytoplasmic vacuolization induced by JB was related to methuosis, cells were treated with JB, and the fusion of vacuoles with lysosomes was examined by confocal microscopy. As illustrated in Figure 1G, vacuoles originated by JB treatment (stained with LY) did not colocalize with the lysosomal marker lysotracker. This suggests that JB disrupts the endosome–lysosome fusion, which is indicative of methuosis [4].

In order to examine further the vacuolization induced by JB, we incubated cells with the JB fluorescent analogue JB-Bodipy (Appendix A), which induced a dose-dependent decrease in cell viability with an IC_50_ of 16 µM (24 h) (Appendix A). A549 cells were treated with JB-Bodipy for a short time (2 h), and examined by phase contrast microscopy. JB-Bodipy induced cellular vacuolization but vacuoles did not contain JB-Bodipy (Appendix A), whose transport across the plasma membrane, as a long-chain base analog, is likely mediated by acyl CoA synthetases [22]. In turn, JB-Bodipy stain was localized in distinct vesicular structures (Appendix A). These results agree with those reported by Rozié et al. [23], who found JB to localize in cytoplasmatic aggregates.

### 2.2. JB-Induced Cytotoxicity Involves Methuosis

JB induced cytotoxicity in A549 cells, with an IC_50_ of 2.05 µM (Appendix A). This result is similar to those reported for other human cancer cell lines, such as HGC27 [10], U2OS [23], HT29 [24], MCF7 [25], A-375 [26], and HeLa [27]. In all the cases, reported IC50 values were in the micromolar to submicromolar range. Some of these cell lines are not mutated in the K-RAS/MAPK pathway, which means that Jaspine B toxicity does not necessarily require hyperactivation of this route.

Next, we investigated the type of cell death induced by JB. As reported with other cell death inducers [2,4,10,11,15,28,29], we observed that the necroptosis inhibitor necrostatin-1 failed to recover the cytotoxicity induced by JB in A459 cells (Appendix A). Therefore, we next investigated the role of apoptosis.

JB has been reported to induce apoptosis in different cancer cell lines [11,12,13]. In our case, we evaluated whether JB induced apoptosis in A459 cells using flow cytometry analysis. While JB showed a poor proapoptotic effect at short (4 h) incubation time, discrete late apoptosis was detected at 24 h treatment (Figure 2A). To investigate the role of apoptosis in JB-induced cytotoxicity, we used the pancaspase inhibitor carbobenzoxy-valyl-alanyl-aspartyl-[O-methyl]-fluoromethylketone (z-VAD). As shown in Figure 2B, z-VAD recovered cell death induced by H_2_O_2_ (a known apoptotic inducer), but it did not recover the viability of cells treated with JB. This result suggests that JB is a poor inducer of apoptosis, and that apoptosis is not the main cause of cell death observed upon JB treatment. To further confirm this observation, we used a genetic approach, using mouse embryonic fibroblasts (MEF) double knockout for both Bcl-2-associated X protein (BAX) and Bcl-2-antagonist/killer 1 (BAK). MEF lacking Bax and Bak cannot undergo apoptosis, as these proteins are essential regulators of the apoptotic signaling [30]. Firstly, we observed that JB induced micropinocytosis-mediated vacuolization in both BAX^+/+^/BAK^+/+^ (wild type) and BAX^−/−^/BAK^−/−^ (knockout) MEF (Appendix A). Interestingly, no difference in cell viability was observed when wild-type and knockout MEF were treated with JB (Figure 2D), suggesting that apoptosis does not mediate JB cytotoxicity. Our results agree with previous works showing that cytotoxicity induced by methuosis is not prevented by apoptosis inhibitors [2,4,7,11,31].

Macroautophagy, hereafter referred to as autophagy, is a highly conserved cellular process characterized by self-degradation of intracellular components that are included in double-membrane vesicles known as autophagosomes [32]. In some cases, autophagy can trigger cell death, a process called autophagy-mediated cell death or cytotoxic autophagy [33]. Interestingly, autophagy has also been described to occur along with methuosis [11]. Therefore, we next studied whether JB induced autophagy in our cells. One of the hallmarks of autophagy is conjugation of the soluble form of the microtubule-associated protein 1 light chain 3 (LC3) with phosphatidylethanolamine and conversion to autophagosomal membrane-associated form (LC3-II), which can be monitored by immunoblot analysis. As shown in (Figure 3A), treatment of A549 cells with JB resulted in rapidly (4 h) elevated LC3-II levels. To demonstrate that JB induced autophagic flux, cells were incubated with the lysosomal inhibitors pepstatin A and E64d, to block the final step of autolysosomal degradation. Incubation with protease inhibitors and E64d further increased LC3-II levels (Figure 3B, 1 vs. 2, 5 vs. 6), indicating that JB induces dynamic autophagy in A549 cells.

We used a pharmacological approach to block the formation of autophagosomes, by inhibiting vacuolar protein sorting 34 class III phosphoinositide 3-kinase (class III PI3K Vps34), an essential protein for cellular autophagy. The incubation of cells with 3-methyladenine resulted in a decrease in JB-induced LC3-II expression levels (Figure 3B, column 5 vs. 7, column 6 vs. 8). However, 3-methyladenine did not recover the loss of cell viability in response to JB (Figure 3C). In order to obtain genetic evidence of whether the JB induces cytotoxic autophagy, we used autophagy-related gene 5 (*ATG5*) knockout MEF (*ATG5*^−/−^). Atg5 protein plays an essential role in the elongation of the phagophore and its maturation into the complete autophagosome; therefore, cells lacking *ATG5* are autophagy-deficient [34]. JB induced macropinocytic vacuoles in both wild-type (*ATG5*^+/+^) and *ATG5*^−/−^ MEF (Figure 3A), but *ATG5* gene deletion had no effect on JB-induced cytotoxicity (Figure 3D).

Finally, to investigate whether vacuoles induced by JB fuse with autophagosomes, we performed fluorescence microscopy colocalization analysis, using a MCF7 cancer cell line that stably overexpresses GFP-tagged LC3-II. TEM analysis revealed the formation of vacuoles by macropinocytosis (Figure 1C) in these cells in response to JB treatment. As shown in Figure 4A, vacuoles and autophagosomes generated in response to JB did not colocalize, indicating that they are different subcellular structures. This was confirmed by confocal microscopy of cells expressing GFP-tagged LC3 and stained with the fluorescent tracer LY. As shown in Figure 4B, most JB-induced vacuoles did not colocalize with autophagosomes (Mander’s test: LY on GFP 0.048; GFP on LY 0.469). Overall, our results suggest that JB does not induce autophagy-mediated cell death, in agreement with previous reports [4,11,15,28].

All in all, our findings strongly suggest that vacuoles formed in response to JB do not merge with lysosomes or autophagosomes, and that JB cytotoxic activity occurs through methuosis. Thus, we propose JB as a methuosis-inducing agent, as it has been proposed for other small molecules, such as 3-(5-methox-y-2-methyl-1H-indol-3-yl)-1-(4-pyridinyl)-2-propene-1-one (MOMIPP) [35] and silmitasertib (CX-4945) [36].

### 2.3. JB Activates AMPK

Recent studies have shown that macropinocytosis contributes to cell growth by stimulating mTORC1 activity [37]. Although in our model macropinocytosis induces cell death rather than contributing to cell growth, we addressed the mTORC1 pathway. To this end, protein kinase B (Akt) and ribosomal protein S6 (S6) phosphorylation was analyzed by immunoblot. JB treatment of A549 cells resulted in a rapid reduction in phosphorylated Akt (30 min) and S6 (at 8 h) levels (Figure 5A, left panels). These results were confirmed in HGC27 human gastric cancer cells, where JB inhibited Akt and S6 phosphorylation after 1 h and 2 h incubation, respectively (Figure 5A, right panels). To investigate the role of the Akt/mTORC1 pathway in JB-induced cell vacuolation, we investigated the effect of everolimus (mTORC1 inhibitor), MK-2206 (Akt inhibitor), and AZD-2014 (mTORC1/2 inhibitor) in A549 cells. None of Akt or mTOR inhibitors induced vacuolization (Figure 5B), in spite of inhibition of mTORC1 (Figure 5C). These results indicate that the Akt/mTORC1 signaling pathway is not involved in the induction of vacuolization in our cells.

Recent work has proposed the AMP-dependent kinase (AMPK) as an upstream inducer of autophagy. AMPK activates the master regulator of autophagy, Atg1/Unc-51-Like Autophagy Activating Kinase 1 (ULK), by two different mechanisms: inhibition of mTORC1 by phosphorylating tuberous sclerosis complex 2 (TSC2) and Raptor proteins, and by direct phosphorylation of Atg1/ULK [38]. Therefore, we first studied whether JB induced AMPK activity in A549 cells. Immunoblot analysis showed that JB induced AMPK phosphorylation (Thr172) as early as upon 1h incubation (Figure 6A). Since JB induced cellular vacuoles within 1–2 h treatment, and mTORC1 activity was unaffected at this time, we next investigated the role of active AMPK in JB-mediated cellular vacuolization. Interestingly, treatment of A459 cells with the AMPK activator phenformin resulted in the induction of vacuoles (Figure 6B), implicating active AMPK as a player in cellular vacuolization. Furthermore, phenformin-induced vacuoles were formed by macropinocytosis (Figure 6B) and did not merge with lysosomes (Figure 6C), as it happened for the vacuoles formed in response to JB. Besides the formation of vacuoles, phenformin induced a clear ER dilation, in agreement with the reported phenformin-induced UPR activation in an AMPK-dependent manner [39].

## 3. Discussion

Methuosis is an emergent type of cell death that has been recently proposed as a target for designing new antitumoral therapies [40]. In this regard, very few small molecules have been reported as methuosis inducers [7,8,9,10,11]. However, most of the protein targets of these compounds remain to be identified. Among them, the inhibition of class III phosphoinositide kinase PIKFYVE [8] triggers vacuolization, a hallmark of methuosis. Additionally, CK2 inhibition by silmitasertib also induces methuosis [10]. Here, we described the natural SL derivative Jaspine B as a compound that triggers vacuolization by micropinocytosis and induces methuosis in cancer cells. Additionally, we provide evidence that JB induces autophagy and apoptosis in cancer cells, but these cellular processes are not involved in the methuosis phenotype induced by JB.

Although the exact signaling pathways by which small compounds induce methuosis are not fully described, implication of Ras (rat sarcoma virus)/Rac (Ras-related C3 botulinum toxin substrate 1) [4,15], and Akt-mTOR [10] pathways have been suggested. Deregulation of macropinocytosis in cancer cells correlates with oncogenic RAS, as RAS-dependent signaling pathways hyperstimulate macropinocytosis in glioblastoma [4], and CD99 triggering induces methuosis of Ewing sarcoma cells through the insulin-like growth factor type 1 receptor (IGF-1R)/RAS/Rac1 signalling [15]. On the other hand, 3-(5-methoxy-2-methyl-1H-indol-3-yl)-1-(4-pyridinyl)-2-propene-1-one (MOMIPP) induces methuosis in glioblastoma and other cancer cell lines by modulating the MAPK/Jun N-Terminal Kinase (JNK) signalling pathway [29]. Interestingly, the CK2 inhibitor silmitasertib promotes methuosis-like cell death associated with massive catastrophic vacuolization, most likely due to the inhibition of the Akt/mTORC1 axis (Akt is a known CK2 target) [10]. In our study, we found that treatment of cancer cells with JB resulted in inhibition of Akt and mTORC1. However, specific inhibition of Akt or mTORC1/mammalian target of rapamycin complex 2 (mTORC2) did not result in vacuolization, discarding the Akt/mTOR pathway as a player in the methuosis phenotype induced by JB in cancer cells. Additionally, here we described that JB induced autophagic flux in cancer cells, but provided genetic evidence that autophagy is not involved in JB-induced macropinocytic vacuoles in these cells.

AMP-activated protein kinase (AMPK) plays an essential role as sensor of energy imbalance in cells, thereby being pivotal for cell survival under various environmental stressors [41]. AMPK is involved in both tumor suppression and tumor-promoting activity depending on distinct cellular contexts. Since AMPK has been shown to display tumor-suppressive roles in certain conditions, several small molecules that activate AMPK have been tested in preclinical models of cancer [42]. Our results suggest that AMPK activation is involved in methuosis induced by JB, in agreement with previous reports. Thus, Kim et al. reported that phosphatase and tensin homolog (PTEN)-deficient prostate cancer cells use macropinocytosis to survive and proliferate under nutrient stress, and that AMPK activation is a general requirement for macropinocytosis of extracellular material induced by PTEN loss [43]. On the other hand, AMPK has been reported to be required for the macropinocytic internalization of ebolavirus [44]. Furthermore, methuosis can be induced by activation of the MAPK/JNK signalling pathway [26]. Our results provide further evidence suggesting that AMPK mediates methuosis in lung cancer cells carrying *KRAS* mutation.

In conclusion, JB induces cancer cell vacuolation and death through a mechanism that cannot be prevented by inhibiting apoptosis, autophagy, or necrosis and it is most likely due to the activation of methuosis. Although more research is needed, we propose that the activation of AMPK would play a critical role in JB-induced cell vacuolization, independently of the PI3K/Akt/mTORC1 axis, which in turn is involved in the autophagic response of cells to JB (Figure 7). 

## 4. Materials and Methods

### 4.1. Reagents and Antibodies

MEM (minimum essential medium), DMEM (Dulbecco’s Modified Eagle Medium), FBS (Fetal bovine serum), nonessential amino acids, trypsin-EDTA, MTT (3-(4,5-dimethylthiazol-2-yl)-2,5-diphenyl tetrazolium bromide), BSA (bovine serum albumin), 5-(N-ethyl- N-isopropyl)amiloride (EIPA), ECL™ Prime Western Blotting Detection Reagent, and 3-methyladenine were from Sigma-Aldrich/Merck. DMEM FluoroBrite, lysotracker red, and Lucifer yellow (LY) were from ThermoFisher Scientific. z-VAD and fumonisin B1 (FB1) were from Enzo Life Sciences. Annexin V-FITC early apoptosis detection kit was from Cell Signaling. Laemmli buffer 4x and 30% acrylamide/Bis 37.5:1 were from Bio-Rad. Lysotracker Red and PDVF membranes were from Roche. Antibodies: β-actin (mouse) was from Sigma; LC3II (rabbit) from MBL; Caspase 3 (rabbit), Akt (rabbit), pAkt (rabbit), AMPK (rabbit), pAMPK (rabbit), S6 (rabbit), and pS6 (rabbit) were from Cell Signaling. HRP-secondary antibody goat anti-Mouse IgG was from Thermo Fisher Scientific (Barcelona, Spain). HRP-secondary antibody goat anti-Rabbit was from Sigma (Fontenay-sous-Bois, France).

### 4.2. Jaspine B and Jaspine B-Bodipy Synthesis

Synthetic pachastrissamine (Jaspine B) was obtained from phytosphingosine using the protocol described by Overkleef et al. [45]. Jaspine B-Bodipy was synthesized by a sequence of reactions, starting with a cross metathesis of two advanced synthones, prepared by procedures previously described. Jaspine B alkene functional moiety was carried out following the methodology described by Jana and Panda [27], and the C10-Bodipy alkene moiety was prepared as described by Saba et al. [46]. The unsaturated isopropylidene-Boc-protected Jaspine B-Bodipy obtained was reduced to the saturated intermediate, and finally deprotected using acidic conditions to afford the wanted fluorophore (Appendix A).

### 4.3. Cell Lines

HGC27 cells (human gastric cancer cells, provided by Prof. Riccardo Ghidoni, University of Milan, Italy) were maintained in MEM supplemented with 10% FBS and 1% nonessential amino acids. Cells were grown without reaching confluence. A549 cells (human lung carcinoma, ATCC). Atg5^−^/^−^ and Atg5^+^/^+^ T-large antigen-transformed MEF cells (provided by Dr. Noboru Mizushima, Tokyo Medical University, Tokyo, Japan), Bax^−^/^−^/Bak^−^/^−^ and Bax^+^/^+^/Bak^+^/^+^ T-large antigen-transformed MEF cells [47] (provided by Stanley J. Korsmeyer, Dana-Farber Cancer Institute, Boston, MA, USA), and MCF7-LC3GFP cells (provided by Dr. Yaowen Wu, Umeå University, Umeå, Sweden) were cultured in DMEM supplemented with 10% FBS. All cells were maintained at 37 °C in 5% CO_2_.

### 4.4. Cell Viability

1 × 10^5^ cells were seeded in 96-well plates and grown for 24 h. Cell viability was examined in triplicate by MTT (3-(4,5-dimethylthiazol-2-yl)-2,5-diphenyltetrazolium bromide) assay. Absorbance was measured at 570 nm using the multi-detection microplate reader BioTek Synergy 2.

### 4.5. Annexin V-FITC Staining

1 × 10^5^ cells were plated in 6-well plates and allowed to grow overnight. After treatment, cells and medium were collected and centrifuged and cellular pellets washed with 50 mM PBS-EDTA 1% BSA. For cell staining, Annexin V-FITC kit (Cell Signaling) was used according to the manufacturer’s instructions. Stained cells were analyzed in a Guava EasyCyte™ flow cytometer (Merck Millipore, Billerica, MA, USA). Data analysis was performed using the Multicycle AV program (Phoenix Flow Systems, San Diego, CA, USA).

### 4.6. Uptake of Fluid-Phase Tracer LY

1 × 10^5^ cells were seeded in 6-well plates or 35 mm glass-bottom dishes and grown overnight. Medium was replaced with 1 mL of fresh medium containing LY (0.5 mg/mL) and cells were incubated with 5 µM JB or 0.05% ethanol (vehicle) for 4 h at 37 °C. Then, Lysotracker (75 nM) was added to each well and left 30 min prior to observation. Medium was removed, cells were washed three times with PBS, and fresh DMEM FluoroBrite was added. Fluorescent images of live cells were taken using a Digital Sight DS-2Mv camera, acquired with Nis Element F 3.0 software or a Zeiss 880 laser scanning confocal microscope and analyzed using Fiji-ImageJ software.

### 4.7. Western Blotting

Cells were lysed in Laemmli sample buffer. Equal amounts of protein (15–30 µg) were loaded and separated on a polyacrylamide/SDS gel and transferred onto a PVDF membrane. Membranes were blocked in 5% milk or 5% BSA in 0.1% TBS-Tween and incubated with the corresponding primary antibody. After washing with 0.1% TBS-Tween, membranes were probed with the correspondent horseradish-peroxidase-conjugated secondary antibody. Finally, protein detection was carried out using ECL reactive and visualized in a LI-COR C-DiGit^®^ blot scanner. Alternatively, AMPK and pAMPK proteins were detected using photographic films (Fuji Medical X-ray film, Fijifilm, Greenwood, SC, USA) in a FUJI PHOTO FPM-100A. Band intensity was quantified by LI-COR Image Studio Lite software.

### 4.8. Phase Contrast and Confocal Microscopy

Cells (0.3 × 10^6^/mL) were seeded in 6-well plates. Phase contrast pictures were taken using a Nikon Eclipse TS100 inverted microscope, connected to a Digital Sight DS-2Mv camera, and acquired with Nis Element F 3.0 software. Confocal microscopy images were obtained in a Zeiss 880 confocal microscope, and analyzed by Fiji-Image J program.

### 4.9. Transmission Electron Microscopy

1 × 10^5^ cells were seeded in 60 mm dishes and grown overnight. Medium was replaced with fresh medium containing JB, phenformin, or ethanol (0.05%). After 4–6 h of treatment, cells were washed 3 times with PBS. Then, cells were fixed in 3% glutaraldehyde in 0.1 M phosphate buffer for 1 h at room temperature. Fixing buffer was removed and fresh fixative buffer was added. Cells were collected by gently scrapping from the Petri dish. Samples were centrifuged (200× *g* for 5 min) and resuspended in the same fixation buffer. Fixing buffer was removed, new fixation buffer was added, and samples were centrifuged again (200× *g* for 5 min) and kept in fixative at 4 °C. Pellets were washed in PB and incubated with 1% OsO4 for 90 min at 4 °C. Then, samples were dehydrated, embedded in Spurr, and sectioned using Leica ultramicrotome (EM VC7, Leica Microsystems, Wetzlar, Germany). Ultrathin sections (50–70 nm) were stained with 2% uranyl acetate for 10 min, a lead-staining solution for 5 min and observed using a transmission electron microscope, JEOL JEM-1010, fitted with a Gatan Orius SC1000 (model 832) digital camera.

## Figures and Tables

**Figure 1 ijms-23-07257-f001:**
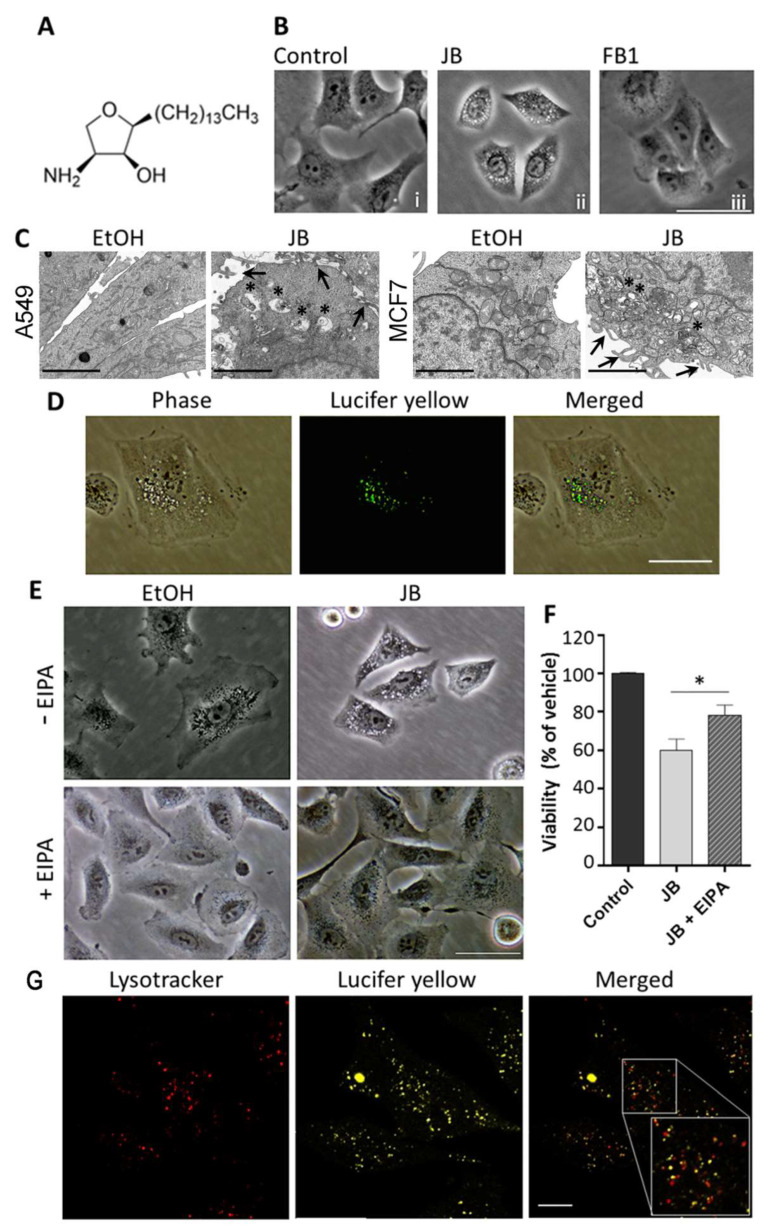
Induction of cell vacuolization by Jaspine B (JB) in A549 cells. (**A**) Chemical structure of JB. (**B**) Phase contrast images of A549 cells treated with 0.05% ethanol for 24 h, 5 µM JB for 4 h, or 50 µM Fumonisin B1 (FB1) for 24 h. The images are representative of three experiments performed in triplicate. Scale bar: 50 µm. (**C**) TEM images of A549 cells treated with 0.05% ethanol for 4 h or 10 µM JB for 5 h and MCF7-LC3GFP cells treated with 0.05% ethanol for 6 h or 10 µM JB for 6 h. Arrows point to membrane extensions at the surface of A549 or MCF7 cells and asterisks show endolysosome accumulation indicating defective endocytic flux. Scale bars: 2 µm. **(D**) Incorporation of Lucifer yellow (LY) in A549 cells incubated with 5 µM JB and 0.5 mg/mL LY for 4 h. Scale bar: 50 µm. (**E**) Phase contrast images of A549 cells pretreated with 25 µM 5-[N-ethyl-N-isopropyl] amiloride (EIPA) (or methanol) for 1 h and then treated with 5 µM JB (or ethanol) for 24 h. The images are representative of two independent experiments performed in triplicate. Scale bar: 50 µm. (**F**) Viability of A549 cells pretreated with 25 µM EIPA (or methanol) for 1 h and then treated with 5 µM JB (or ethanol) for 24 h, as assessed with 3-(4,5-dimethylthiazol-2-yl)-2,5-diphenyl tetrazolium bromide (MTT). Control cells were treated with vehicles (without EIPA and without JB). Results are the mean ± SD of two experiments in triplicate. * *p* < 0.0005. (**G**) A549 cells were treated with 5 µM JB and 0.5 mg/mL LY for 4 h. An amount of 75 nM LT was added 0.5 h prior to visualization. Scale bar: 10 µm.

**Figure 2 ijms-23-07257-f002:**
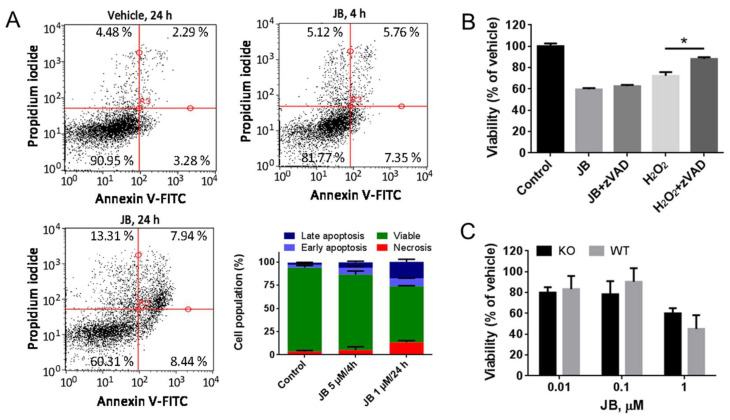
Analysis of apoptosis induced by Jaspine B (JB) in A549 cells. (**A**) A549 cells were incubated with 0.05% EtOH for 24 h (control), 5 µM JB for 4 h or 1 µM JB for 24 h and then stained with annexin V (AV)/propidium iodide (PI). Fluorescence was analyzed by flow cytometry. Data were obtained from 2 experiments in triplicate. (**B**) Determination of cell viability by MTT assay in A549 cells treated for 24 h with 1 µM JB in the presence or absence of 100 µM carbobenzoxy-valyl-alanyl-aspartyl-[O-methyl]-fluoromethylketone (z-VAD) or 10 µM H_2_O_2_ in the presence or absence of 100 µM z-VAD. Results are the mean ± SD for two experiments performed in triplicate and are expressed as the percentage of the viability compared to the control. * *p* < 0.005. (**C**) Determination of cell viability by MTT assay of mouse embryonic fibroblasts (MEF) double knockouts (Bax/Bak KO) for Bcl-2-associated X protein (Bax) and Bcl-2-antagonist/killer 1 (Bak) and MEF wild-type (Bax/Bak WT) incubated with different concentrations of JB for 24 h. Results are the mean ± SD of two experiments in triplicate.

**Figure 3 ijms-23-07257-f003:**
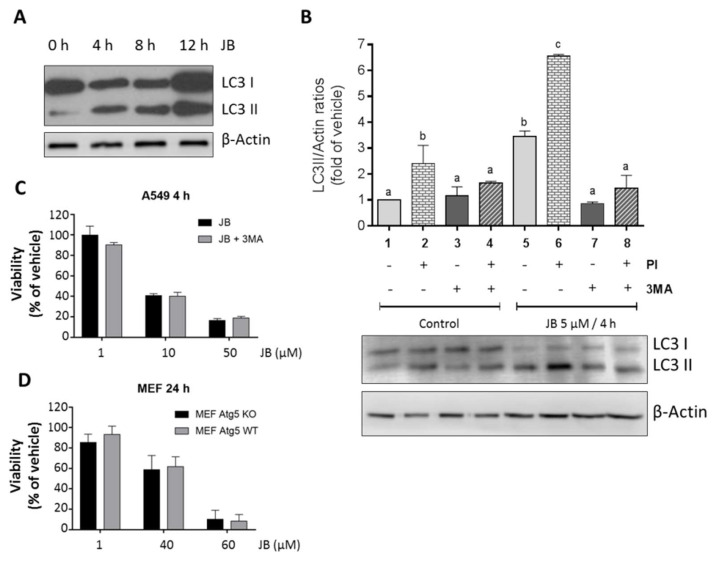
Analysis of autophagy induced by Jaspine B (JB) in A549 cells. (**A**) Analysis of microtubule-associated protein 1 *light chain* II (LC3II) expression by Western blot in A549 cells treated with JB 5 µM at different times for up to 12 h. (**B**) Analysis of LC3II expression by Western blot in A549 cells pretreated or not for 2 h with protease inhibitors (PI) (10 µg/mL E-64-D and 5 µg/mL pepstatin A) and/or for 0.5 h with 5 mM 3-methyladenine (3MA) and then treated with JB (5 µM, 4 h) or vehicle (0.05% EtOH). Results are representative of three experiments. (**C**) Determination of cell viability with 3-(4,5-dimethylthiazol- 2-yl)-2,5-diphenyl tetrazolium bromide (MTT) in A549 preincubated or not with 5 mM 3-methyladenine for 0.5 h and treated with JB at different concentrations for 4 h. Results are the mean ± SD of two experiments in triplicate. (**D**) Determination of cell viability by MTT assay in autophagy-related gene 5 (Atg5)^−/−^ (Atg5 KO) and Atg5^+/+^ (Atg5 WT) mouse embryonic fibroblasts (MEF) incubated with different concentrations of JB for 24 h. Data (+/−SD) correspond to three experiments in triplicate.

**Figure 4 ijms-23-07257-f004:**
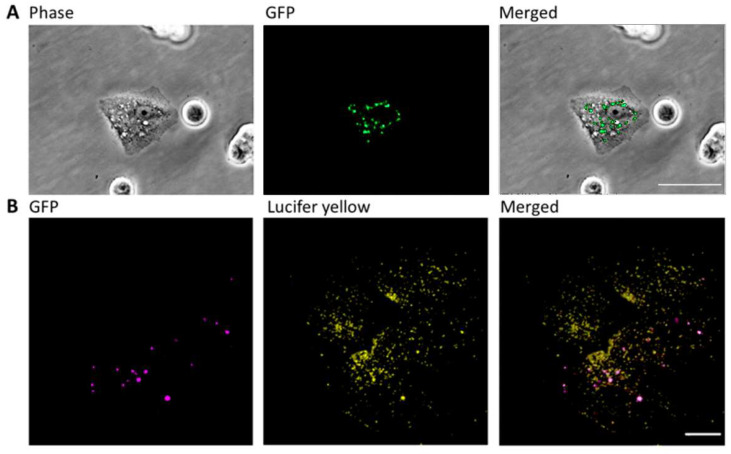
Analysis of vacuoles and autophagosomes in MCF7 cells. (**A**) MCF7 cells stably transfected with GFP-tagged microtubule-associated protein 1 light chain II (LC3II) were incubated with JB (10 µM, 2 h). Vacuoles were observed under a phase microscopy and autophagosomes, labeled with GFP, were analyzed by fluorescence microscopy. Scale bar: 50 µm. (**B**) MCF7 cells stably transfected with GFP-tagged LC3II were incubated with 10 µM JB and 0.5 mg/mL Lucifer yellow (LY) for 2 h and analyzed by confocal microscopy. Images are representative of at least two experiments. Scale bar: 10 µm.

**Figure 5 ijms-23-07257-f005:**
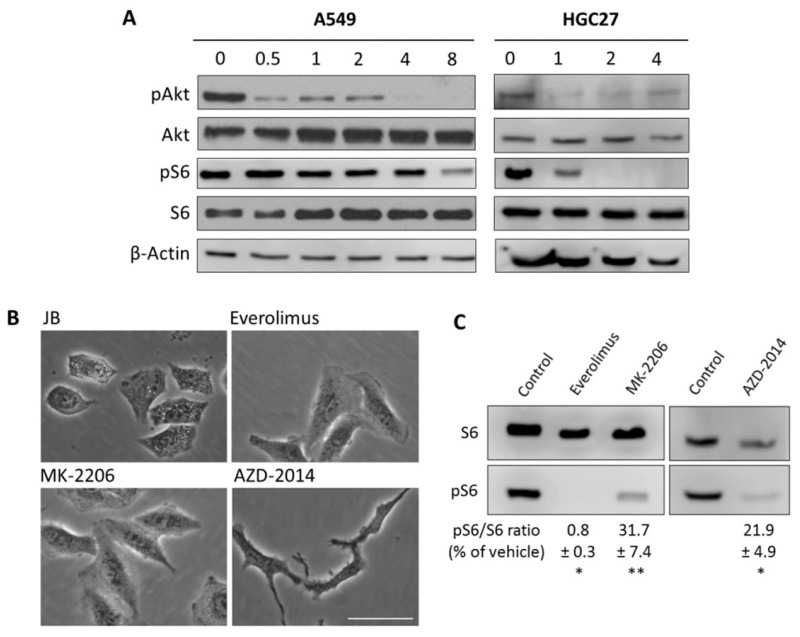
(**A**) A549 and HGC27 cells treated with vehicle (0, 0.05% EtOH, 24 h) or 5 µM JB at the indicated times (h) were lysed, and the levels of total and phosphorylated Akt and S6 proteins were analyzed by immunoblotting. β-Actin was used as loading control. Results are representative of two to three independent experiments. (**B**) A549 cells were treated for 24 h with 1 µM everolimus, 10 µM MK-2206, or 100 nM AZD-2014 and compared to 4 h with 5 µM JB. Phase contrast images are representative of two experiments performed in triplicate. Scale bar: 50 µm. (**C**) Protein extracts were analyzed by immunoblotting for total ribosomal protein S6 and phospho-S6. Numbers denote quantification of levels of phospho-S6 normalized by S6 protein, expressed as percentage of vehicle. Results are representative of two experiments. * *p* < 0.01, ** *p* < 0.0001.

**Figure 6 ijms-23-07257-f006:**
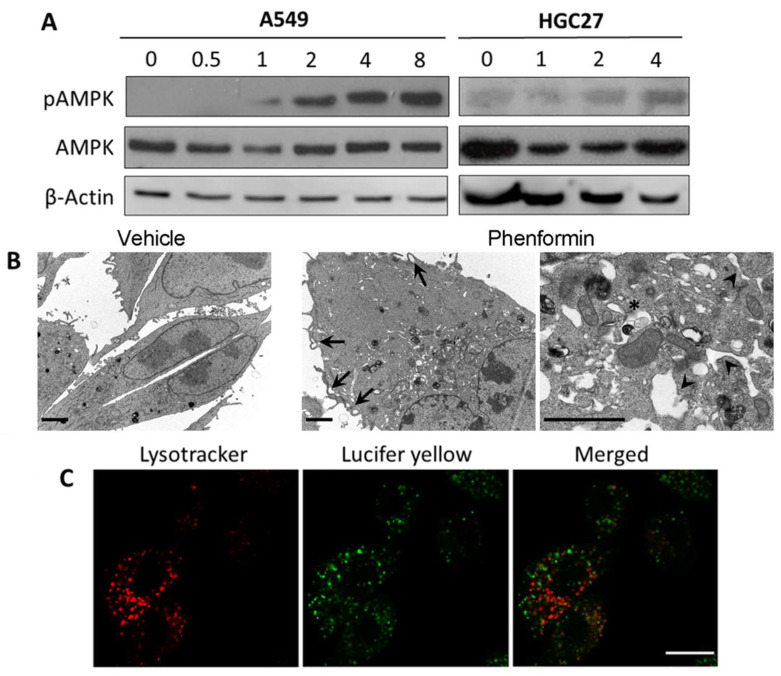
(**A**) A549 and HeLa cells treated with vehicle (0, 0.05% EtOH 24 h) or 5 µM JB at the indicated times (h) were lysed, and the levels of total and phosphorylated 5′ AMP-activated protein kinase (AMPK) were analyzed by immunoblotting. β-Actin was used as loading control. Results are representative of two to three independent experiments. (**B**) A549 cells were treated with 0.05% ethanol or 10 mM phenformin for 4 h. Arrows show membrane extensions, the asterisk shows endolysosome accumulation indicating defective endocytic flux, and the arrowhead, dilated ER. Scale bars: 2 µm. (**C**) A549 cells treated with 10 mM phenformin and 0.5 mg/mL of LY for 4 h. 75 nM lysotracker was added 0.5 h prior to visualization. Cells were observed under confocal microscopy. Scale bar: 10 µm. Images are representative of two experiments in duplicate.

**Figure 7 ijms-23-07257-f007:**
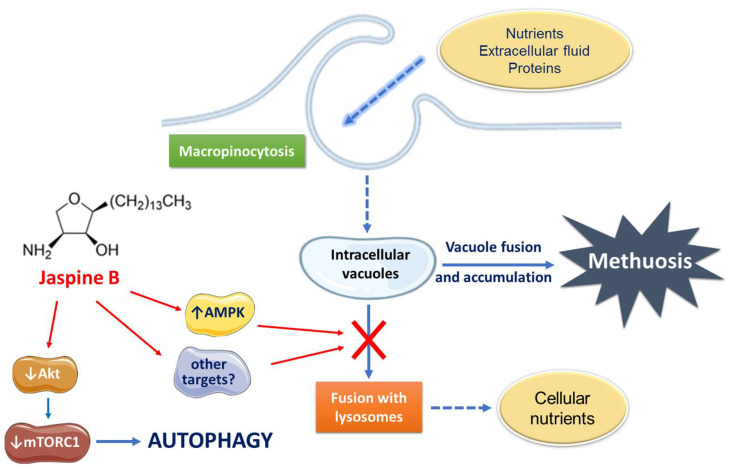
Proposed mode of action of Jaspine B in lung adenocarcinoma A549 and gastric cancer HGC-27 cells.

## Data Availability

Data supporting reported results can be provided upon request.

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
