# Peer review of "Methuosis Contributes to Jaspine-B-Induced Cell Death"

_ijms, 2022, doi:10.3390/ijms23137257_

Round 1

Reviewer 1 Report

Major comments:

In this manuscript Bielsa et al. build up on their previous studies to provide more insights about the vacuolization and cell death induced by the natural small molecule Jaspine B. Some interesting observations are reported, however many experiments do not provide a definitive conclusion or rely on a single inhibitor (e.g. EIPA) to draw conclusions. I recommend tuning down some of the claims to better fit to the actual data as described below:

1/ The authors do not show that AMPK activation is mediating the induction of methuosis/vacuolization by Jaspine B treatment. To show this they should perform a co-treatment of cells with Jaspine B and an AMPK inhibitor (e.g. Dorsomorphin) and monitor if Jaspine B fails to induce methuosis/vacuolization under these conditions. Instead they show that a known AMPK activator induces vacuolization, similarly to Jaspine B, which is a simple correlation. Therefore, the authors should tune down their findings: In the abstract, from the data presented, they cannot claim that “activation of 5' AMP-activated protein kinase (AMPK) plays a critical role in Jaspine B-induced vacuolation”. “…could be involved in Jaspine B-induced vacuolization” would be more accurate.

2/ The title of the manuscript currently suggests that Jaspine B-induced methuosis depends on AMPK, which again is not supported (nor disproved) by the provided data. A more accurate title would be “Jaspine B induces cell death through methuosis” (or “Methuosis contributes to Jaspine B-induced cell death”), which was not previously reported, leaving the observation of AMPK activation upon Jaspine B treatment for the abstract and discussion.

Figure A3: A high concentration of Jaspine B-Bodipy is used (as compared to Jaspine B): the authors should provide a viability curve of Jaspine B-Bodipy to explain why this concentration was chosen.

Page 4: For completion, the authors should also cite U2OS cells (ref 23) in the list of cells in which Jaspine B displays an IC50 in the micromolar to sub-micromolar range. They should also indicate that some of these cell lines are not mutated in the K-RAS/MAPK pathway, which means that the Jaspine B toxicity does not necessarily require hyperactivation of this pathway.

Minor comments:

Figure 1F: It is unclear whether a control with EIPA alone was performed. The legend should be clarified to explain what is the control condition on the graph.

Figure 2A: The axis should be properly labeled. The % of cells in each quadrant should be indicated on the graph.

Figure 6: in the legend, a space is lacking between “or” and “10 mM”.

Figure A5 legend: "Vacuoles induced by Jaspine B (JB) in different cell lines are originated by macropinocytosis." Should indicate “in different murine cell lines”.

Author Response

Point-by-point answer to reviewer 1.

1/ The authors do not show that AMPK activation is mediating the induction of methuosis/vacuolization by Jaspine B treatment. To show this they should perform a co-treatment of cells with Jaspine B and an AMPK inhibitor (e.g. Dorsomorphin) and monitor if Jaspine B fails to induce methuosis/vacuolization under these conditions. Instead they show that a known AMPK activator induces vacuolization, similarly to Jaspine B, which is a simple correlation. Therefore, the authors should tune down their findings: In the abstract, from the data presented, they cannot claim that “activation of 5' AMP-activated protein kinase (AMPK) plays a critical role in Jaspine B-induced vacuolation”. “…could be involved in Jaspine B-induced vacuolization” would be more accurate.

ANSWER.‑ We thank the referee for this comment, to which we definitely agree. In the Abstract, the sentence “activation of 5' AMP-activated protein kinase (AMPK) plays a critical role in Jaspine B-induced vacuolation” has been replaced by “activation of 5' AMP-activated protein kinase (AMPK) could be involved in Jaspine B-induced vacuolization”

2/ The title of the manuscript currently suggests that Jaspine B-induced methuosis depends on AMPK, which again is not supported (nor disproved) by the provided data. A more accurate title would be “Jaspine B induces cell death through methuosis” (or “Methuosis contributes to Jaspine B-induced cell death”), which was not previously reported, leaving the observation of AMPK activation upon Jaspine B treatment for the abstract and discussion.

ANSWER.‑ We thank the referee for this comment, to which we definitely agree. The original title (“Jaspine B Induces Methuosis through Activation of AMPK” has been changed into “Methuosis contributes to Jaspine B-induced cell death”

Figure A3: A high concentration of Jaspine B-Bodipy is used (as compared to Jaspine B): the authors should provide a viability curve of Jaspine B-Bodipy to explain why this concentration was chosen.

ANSWER.‑ We thank the referee for this comment. We have added the viability curve for Jaspine B-Bodipy as Figure A1,B. The legend has been modified to include the new panel.

Page 4: For completion, the authors should also cite U2OS cells (ref 23) in the list of cells in which Jaspine B displays an IC50 in the micromolar to sub-micromolar range. They should also indicate that some of these cell lines are not mutated in the K-RAS/MAPK pathway, which means that the Jaspine B toxicity does not necessarily require hyperactivation of this pathway.

ANSWER.‑ We acknowledge the referee for these comments. We have modified the first sentences of section 2.2. (“JB induced cytotoxicity involves methuosis”), which now read: “JB induced cytotoxicity in A549 cells, with an IC50 of 2.05 µM (Figure A1). This result is similar to those reported for other human cancer cell lines, such as HGC27 [10], U2OS [23], HT29 [24], MCF7 [25], A-375 [26] and HeLa [27]. In all the cases, reported IC50 values were in the micromolar to sub-micromolar range. Some of these cell lines are not mutated in the K-RAS/MAPK pathway, which means that Jaspine B toxicity does not necessarily require hyperactivation of this route.”

Minor comments:

Figure 1F: It is unclear whether a control with EIPA alone was performed. The legend should be clarified to explain what is the control condition on the graph.   

ANSWER.‑ We thank the referee for this comments. In preliminary experiments, we found that treatment of A549 cells with 25 µM EIPA for 1 h did not modify the viability of A549 cells further treated with ethanol for 24 h. In all subsequent experiments, control cells were incubated with vehicle only. We have clarified this in the legend. “….as assessed with 3-(4,5-dimethylthiazol-2-yl)-2,5-diphenyl tetrazolium bromide (MTT). Control cells were treated with vehicles (without EIPA and without JB). Results….

Figure 2A: The axis should be properly labeled. The % of cells in each quadrant should be indicated on the graph.

ANSWER.‑ These changes have been made.

Figure 6: in the legend, a space is lacking between “or” and “10 mM”.

ANSWER.‑ This typo has been corrected.

Figure A5 legend: "Vacuoles induced by Jaspine B (JB) in different cell lines are originated by macropinocytosis." Should indicate “in different murine cell lines”.

ANSWER.‑ The suggested change has been made

Reviewer 2 Report

This work, entitled "Jaspine B Induces Methuosis through Activation of AMPK" from the Casas laboratory, is well-conceived and clearly executed. Natural compounds are of interest because of their potential use in cancer, and Jaspine B (JB), a sphingolipid (SL) derivative, is especially interesting due to it's ability to initiate methuosis, which is independent of alterations it induces in SL metabolism. Specifically, this work focuses on the mechanism of action and signalling pathways involved in JB cytotoxicity in a lung adenocarcinoma model, A549 cells. To this end, the authors have provided evidence of JB-induced cytoplasmic vacuolation (using phase contrast, confocal and transmission electron microscopy) and shown that activation of 5' AMP-activated protein kinase (AMPK) plays a star role in JB-induced vacuolation/cytotoxicity. The work is well-controlled and the methods are sound. The Introduction and Discussion are clear, providing insight and relevance. Importantly, methuosis is a novel, emergent type of cell death recently proposed as a target for development of new anticancer therapies. A suggestion to help round-out the work would be to add a figure (cartoon) showing mechanism and signalling involved in JB action in A549 cells/and perhaps in their gastric cancer model.

Round 2

Reviewer 1 Report

The authors addressed all my concerns through this revised version, which is now suitable for publication.

This manuscript is a resubmission of an earlier submission. The following is a list of the peer review reports and author responses from that submission.